# Altered Differential Expression of Genes and microRNAs Related to Adhesion and Apoptosis Pathways in Patients with Different Phenotypes of Endometriosis

**DOI:** 10.3390/ijms24054434

**Published:** 2023-02-23

**Authors:** Luana Grupioni Lourenço Antonio, Juliana Meola, Ana Carolina Japur de Sá Rosa-e-Silva, Antonio Alberto Nogueira, Francisco José Candido dos Reis, Omero Benedicto Poli-Neto, Julio César Rosa-e-Silva

**Affiliations:** Ribeirão Preto Medical School, University of São Paulo, 3900 Bandeirantes Av., Ribeirão Preto 14049-900, SP, Brazil

**Keywords:** endometriosis, genes, microRNAs, *MAPK1*, *CAPN2*

## Abstract

We aim to investigate the expression of genes (*MAPK1* and *CAPN2*) and microRNAs (miR-30a-5p, miR-7-5p, miR-143-3p, and miR-93-5p) involved in adhesion and apoptosis pathways in superficial peritoneal endometriosis (SE), deep infiltrating endometriosis (DE), and ovarian endometrioma (OE), and to evaluate whether these lesions share the same pathophysiological mechanisms. We used samples of SE (*n* = 10), DE (*n* = 10), and OE (*n* = 10), and endometrial biopsies of these respective patients affected with endometriosis under treatment at a tertiary University Hospital. Endometrial biopsies collected in the tubal ligation procedure from women without endometriosis comprised the control group (*n* = 10). Quantitative real-time polymerase chain reaction was performed. The expression of *MAPK1* (*p* < 0.0001), miR-93-5p (*p* = 0.0168), and miR-7-5p (*p* = 0.0006) was significantly lower in the SE group than in the DE and OE groups. The expression of miR-30a (*p* = 0.0018) and miR-93 (*p* = 0.0052) was significantly upregulated in the eutopic endometrium of women with endometriosis compared to the controls. MiR-143 (*p* = 0.0225) expression also showed a statistical difference between the eutopic endometrium of women with endometriosis and the control group. In summary, SE showed lower pro-survival gene expression and miRNAs involved in this pathway, indicating that this phenotype has a different pathophysiological mechanism compared to DE and OE.

## 1. Introduction

Endometriosis is a benign gynecological disease characterized by the presence and growth of endometrial tissue outside the uterine cavity and myometrium, commonly found in the anterior and posterior compartments of the pelvic cavity, in the pelvic peritoneum, ovaries, rectovaginal septum, bladder, and intestine. It mainly affects women of reproductive age (5% to 10% of women in this phase) and is associated with pelvic pain, dysmenorrhea, dyspareunia, constipation, dysuria, changes in bowel function, and, often, infertility [1]. However, clinical presentation is considerably variable, and none of these symptoms are specific for the disease, making its diagnosis difficult [2].

In addition, endometriosis can significantly affect a woman’s personal, as well as intimate and professional aspects of life. Some studies have shown a higher incidence of depressive symptoms, anxiety, stress, and lower quality of life in women with endometriosis compared to women without endometriosis. Furthermore, a correlation was found between depression/infertility and pain/quality of life [3,4].

Although the disease has heterogeneous presentation, the three main different phenotypes that are currently recognized are ovarian endometrioma, superficial peritoneal endometriosis, and deep infiltrating endometriosis [5]. Superficial endometriosis is thus classified as the presence of small lesions, measuring between 1 mm and 3 mm, with foci generally implanted in the peritoneum and rarely affecting organs. Endometrioma, in turn, is characterized by the presence of cysts in the ovary that are filled with a typical chocolatey liquid. Meanwhile, deep endometriosis presents lesions of at least 5 mm in depth and can be located in several organs, including the peritoneum, bladder, and intestine, the latter being the most advanced form of the disease [6]. These phenotypes may represent three clinically separate disease entities with different pathogenesis. Most deep infiltrating endometriosis lesions present with other forms of endometriosis, and about half deep infiltrating endometriosis lesions present with ovarian endometrioma. It is estimated that deep infiltrating endometriosis can affect 20% of women with superficial peritoneal endometriosis [7]. Clinical data suggest a clear distinction in terms of diagnosis, treatment, and follow-up regarding these three types of phenotypes [5]. However, most pathogenetic research on endometriosis has been conducted mixing the three phenotypic manifestations. Therefore, a better molecular understanding of these lesions is needed in order to provide efficient and specific management for each subgroup [8].

The etiology of endometriosis is complex and multifactorial, where several not fully confirmed theories describe its pathogenesis [9]. Retrograde menstruation is considered an important source of endometrial deposits, although other factors are necessary to promote cell survival, proliferation, formation, and maintenance of endometriotic lesions [1]. Today, it is recognized that alterations in the peritoneal microenvironment must also occur; therefore, the following processes are essential: escaping from immune system surveillance [9,10]; changes in local concentrations of hormones [11] and inflammatory mediators [12]; cell adhesion [13]; tissue invasion [14]; evading apoptosis [15]; angiogenesis [16]; and ectopic cell proliferation [17]. More recently, the possibility that endometriosis is an epigenetic disease was proposed, with modifications in the methylation of DNA and histones and alterations in the expression of non-coding RNAs, among the latter, microRNAs (miRNAs) [18].

Specific changes in endometrial and peritoneal cell adhesion molecules seem to facilitate the binding of endometrial menstrual reflux in ectopic sites, namely the integrins α2β1, α3β1, α4β1, α5β1, and E-cadherin [13]. In this context, most studies have evaluated the expression of integrins in the eutopic endometrium [19]. Studies have shown that patients with endometriosis present reduced susceptibility to apoptosis in endometrial cells released during menstruation, thereby facilitating survival and ectopic implantation [15].

miRNAs are important regulators of gene expression, acting at the post-transcriptional level, through the induction of mRNA degradation or by blocking protein synthesis [20]. They seem to be potent regulators of gene expression in endometriosis, participating in important cellular events that trigger the development of the disease [21]. Several studies have shown that the expression of miRNAs is altered in the eutopic endometrium [22,23,24], in ectopic and eutopic endometrial tissues [21,24,25], and in circulating miRNAs in women with endometriosis when compared to healthy subjects [26,27,28,29,30].

Most studies on miRNA and endometriosis have been carried out comparing the endometrium of women with and without endometriosis, encountering, in many cases, differential expression and, therefore, implicating miRNAs as potential biomarkers of this condition [31]. Although obtaining endometrial tissue presents an invasive aspect, the advantage of its use is that it can be accessed by biopsy without the need for anesthesia [32]. However, a panel with good specificity and sensitivity has not been found so far [33,34].

Unraveling the significance of miRNAs in endometriosis will pave the way for a better understanding of the pathophysiology of this disease and new diagnostic tests, as well as identify new therapeutic targets and treatment approaches that have the potential to improve the clinical options for women with this disabling condition. However, despite the increase in research on the subject, to date, the utility of miRNAs for this purpose has not been specifically analyzed [33]. Thus, the aim of the present study was to investigate the expression of the genes *(MAPK1* and *CAPN2*) and microRNAs (miR-30a-5p, miR-7-5p, miR-143-3p, and miR-93-5p) involved in the adhesion and apoptosis pathways in superficial peritoneal endometriosis (SE), deep infiltrating endometriosis (DE), and ovarian endometrioma (OE), and to evaluate whether these lesions share the same pathophysiological mechanisms.

## 2. Results

The demographic variables of the patients did not differ significantly among the groups (*p* > 0.05) (Table 1).

The expression of genes and microRNAs among the ectopic implants is illustrated in Figure 1. The expression of *MAPK1* (*p* < 0.0001) and miR-7-5p (*p* = 0.0006) was significantly lower in superficial lesions compared to deep endometriosis and ovarian endometrioma. Meanwhile, the expression of miR-93-5p showed to be different between superficial and deep lesions (*p* = 0.0168). The relative expression of *CAPN2*, miR-143-3p, and miR-30a-5p did not differ significantly (*p* > 0.05) among the different lesion types.

Gene and microRNA expression were compared between the control eutopic endometrium and the eutopic endometrium of women with endometriosis. The expression of miR-30a and miR-93 was significantly (*p* = 0.0018 and *p* = 0.0052, respectively) upregulated in the eutopic endometrium of women with endometriosis compared to the controls. The expression of miR-143 was also statistically different between groups (*p* = 0.0225); however, it was downregulated in patients with endometriosis. There was no significant difference regarding the expression of *MAPK1*, *CAPN2*, and miR-7 (Figure 2).

The expression of genes and miRNAs between the eutopic endometrium of women with different types of lesions and the eutopic endometrium of the controls were compared, as shown in Figure 3. The expression of miR-30a was significantly lower in the control eutopic endometrium than in the endometrium of women with deep infiltrating lesions (*p* = 0.0194) and ovarian endometrioma (*p* = 0.0208). Moreover, the eutopic endometrium of the control group showed reduced expression of miR-93 when compared to the eutopic endometrium of patients with superficial peritoneal lesions (*p* = 0.0139).

The expression of genes and microRNAs in the ectopic implants was compared with the expression of the corresponding eutopic endometrium (within the same woman). The expression of *MAPK1* (*p* < 0.0001) and miR-93 (*p* = 0.0021) was significantly lower in the lesions when compared to the corresponding eutopic endometrium (Figure 4).

## 3. Discussion

The different theories that attempt to explain the etiology of endometriosis seem to complement each other, suggesting that the disease has a multifactorial origin. However, the exact mechanisms that promote and favor the survival and implantation of endometriotic foci in ectopic sites have not yet been precisely clarified. Thus, further studies aimed at investigating the etiology of endometriosis are still needed. The knowledge of its etiology will make it easier to develop new diagnostic tests, as well as identify new therapeutic targets and treatment approaches that have the potential to improve the clinical options for women with this disabling condition. Furthermore, it is necessary to distinguish the pathophysiological mechanisms in the different phenotypes of endometriosis in order to contribute to the individualized care of these patients.

It was suggested long ago that the phenotypes ovarian endometrioma, superficial peritoneal endometriosis, and deep infiltrating endometriosis may represent three clinically separate disease entities with different pathogenesis; however, different types of endometriosis very often coexist in one patient [7]. In our experiments, only one type of lesion from each patient was collected.

The results obtained in the present study demonstrated a significantly lower expression of *MAPK1* in superficial lesions compared to deep endometriosis and ovarian endometrioma. Moreover, *MAPK1* expression was significantly lower in the lesions when compared to the corresponding eutopic endometrium. *MAPK1* are extracellular signal-regulated kinases (ERKs), such as sex hormones and inflammatory factors, that play an important role in many cellular reactions [35]. Several studies have shown this gene directly participating in the regulation of endometriosis pathophysiology, such that the MAPK pathway was activated in ectopic and eutopic endometrial cells of patients with endometriosis [36]. Research using *MAPK1* inhibitors has evidenced anti-inflammatory, anti-proliferative, anti-angiogenic, and apoptotic effects and reduced adhesion and migration [36,37,38,39,40,41,42]. Thus, our results demonstrate a likely greater activation of these pathways in deep lesions and ovarian endometrioma, and that the superficial endometriosis phenotype probably has a different pathophysiological mechanism.

The signaling cascades of the MAPK pathway participate in cell survival mechanisms, providing signals that fuel cell cycle progression and affect the transcription factors that regulate apoptosis. The latter can activate Bcl-2 or inactivate caspase-9, thus generating an anti-apoptotic signal [36]. Li et al. (2013) showed that the ability of human primary cell cultures from eutopic endometrial stroma to adhere to Collagen IV and Fibronectin is MAPK-dependent. They found that U0126 (a MEK-targeted MAPK inhibitor) affected endometriotic stromal cell adhesion and invasion in vitro [40].

Ngo et al. (2010) compared ectopic and eutopic endometrial cells from biopsies of patients with and without endometriosis, and observed that the MAPK pathway was activated in ectopic and eutopic endometrial cells from patients with endometriosis, as evidenced by a significantly higher pERK/ERK ratio in these patients compared to the control group [43]. Furthermore, the increased proliferation and survival of eutopic endometrial cells from patients with endometriosis, compared to healthy women, has been correlated with abnormal levels of activation of the MAPK signal pathway [44]. However, according to our results, no difference was found in *MAPK1* expression in the eutopic endometrium of women with superficial peritoneal endometriosis, deep infiltrating endometriosis, ovarian endometrioma, and control eutopic endometrium.

According to a recent meta-analysis, there are consistently upregulated and downregulated miRNAs in ectopic foci compared to the eutopic endometrium of healthy women [34]. MiR-93-5p and miR-143-3p regulate *MAPK1* gene expression. Research on endometriosis has already been carried out for both miRNAs. The results of the present study demonstrated an upregulation of miR-93-5p expression in the eutopic endometrium of women with endometriosis when compared to the controls, the difference being between control endometrium *versus* endometrium in women with superficial peritoneal lesions. Furthermore, the expression of miR-93 was different between superficial and deep lesions, with greater expression in the latter. This result contrasts with a previous study, in which miR-93 showed to be underexpressed in the ectopic endometrium of patients with endometriosis compared to the peritoneal tissue of patients without the disease, causing increased expression of *MMP3* and *VEGFA*, thus stimulating the proliferation, migration, and invasive capacity of endometrial stromal cells [45]. Such difference between results is probably due to the different tissues analyzed in the comparison between endometriosis and the control.

MiR-143-3p is involved in cell proliferation, apoptosis, adhesion, invasion, and other cellular processes [46]. The transfection of a miR-143 mimetic into HL-60 myelocytic leukemia cells remarkably suppressed *MAPK1* expression, inhibiting cell proliferation and inducing apoptosis [47]. Chang et al. (2017) found that miR-143 inhibited proliferation, migration, and invasion and promoted apoptosis in endometrial cancer cells by suppressing *MAPK1* [48]. Other studies have demonstrated the role of miR-143 as a tumor suppressor, reducing proliferation and adhesion and increasing apoptosis [49,50]. Previous studies reported that miR-143-3p was markedly dysregulated in endometriosis and was found to be overexpressed in the serum and tissue of affected women when compared to the controls [51,52,53].

In our study, downregulation of miR-143 expression was observed in the eutopic endometrium of women with endometriosis compared to the control endometrium. Conversely, in a recent study, the expression of miR-143-3p was upregulated in endometriotic stromal cells from women with endometriosis when compared to eutopic endometrial tissues obtained from women without endometriosis. However, functionally, the overexpression of miR-143-3p suppressed the proliferation and invasion of endometriotic stromal cells, thus inhibiting the progression of the disease [53]. Therefore, a lower expression of miR-143 in the endometrium could promote a greater potential to proliferate and invade, corroborating our results.

Previous studies showed that miR-143 was overexpressed in ectopic endometrial tissues when compared to eutopic endometrial tissues [21,54]. The same result was obtained herein in the group presenting deep lesions. Similar to our findings, in serous ovarian carcinoma, there was an overexpression of miR-93 and an underexpression of miR-143 compared to benign lesions, thus correlating with lower patient survival [55].

Genes from the CAPN family are calcium-activated neutral proteases that have been reported to regulate focal adhesion, cytoskeletal remodeling, and apoptosis [56]. The *CAPN5* gene was found to be underexpressed in eutopic endometrial biopsies of women with endometriosis when compared to controls without the disease [57]. On the other hand, another study reported an increased expression of *CAPN7* in the eutopic endometrium and endometrial stromal cells of women diagnosed with endometriosis. The authors proposed, based on functional tests, that this gene promotes the migration and invasion of human endometrial stromal cells through the regulation of matrix metalloproteinase 2 (MMP-2) [58]. However, in the present study, we did not find differences among the different types of lesions, nor between the control and endometriosis groups regarding the expression of the *CAPN2* gene.

According to the miRwalk program, miR-30a-5p and miR-7-5p are regulators of *CAPN2* gene expression. The expression of miR-7 was significantly lower in the superficial lesion group than in the deep endometriosis and ovarian endometrioma groups. MiR-7 has been reported as a regulator of *MMP-2* and *MMP-9* expression, acting in the invasion and proliferation of human colon cancer by directing the expression of the focal adhesion kinase [59]. Moreover, the expression of *MMP-2* and *MMP-9* has been shown to be increased in women with endometriosis when compared to controls [60]. Therefore, this pathway may be more activated in the deep endometriosis and ovarian endometrioma phenotypes.

MiR-7 was found to be underexpressed in cervical cancer tissues compared to the corresponding normal adjacent cervical tissues. Furthermore, the expression of miR-7 in metastatic cervical cancer was significantly lower compared to cancer without metastasis. The overexpression of miR-7 by transfection inhibited the migration and invasion of cervical cancer cells by suppressing the expression of *FAK* (focal adhesion kinase), an important adhesion kinase that contributes to extracellular matrix integrin signaling, cell motility, proliferation, and survival [61]. MiR-7 overexpression in cervical cancer cell lines (HeLa and C-33A) also suppressed cell viability and promoted apoptosis, while its inhibition promoted opposite effects [62].

The expression of miR-30a was upregulated in the eutopic endometrium of women with endometriosis compared to the controls, with the difference in its expression occurring between the control *versus* deep infiltrating lesions and ovarian endometrioma. MiR-30a downregulates the expression of β3-integrin, modulating cell adhesion and invasion, interrupting the *MAPK1* pathway in triple-negative breast cancer [63]. Corroborating our results, some studies have reported reduced expression of β3-integrin in the endometrium of women with endometriosis [16].

In the present study, we conducted comparisons between eutopic endometrium and ectopic lesions within cases, assuming that the differential expression of genes and miRNAs in the affected women would reflect disease processes as opposed to individual differences in gene expression and regulation. In this comparison, we found a difference in the expression of miR-93, which, in the group of women with superficial endometriosis, exhibited upregulated expression in the eutopic endometrium compared to the ectopic lesions. Meanwhile, the expression of miR-143 in the group of women with deep endometriosis was downregulated in the eutopic endometrium when compared to the ectopic lesions.

It is well-known that the types of endometriotic lesions are biochemically distinct and, therefore, it is hypothesized that miRNAs are differentially expressed in superficial peritoneal endometriosis, deep infiltrating endometriosis, and endometrioma. In this context, a major criticism in studies carried out in this area is the lack of molecular evaluations of endometriosis in the different phenotypes of the disease. So far, few studies have made such distinction. Haikalis et al. (2018) separately analyzed miRNA expression in the three types of lesions. In their study, 15 endometrioma samples, 11 superficial lesion samples, and 10 deep lesion samples were used, and the expression levels of miR-9, miR-21, miR-424, miR-10a, miR-10b, and miR-204 were evaluated by qPCR. The expression of miR-21 and miR-424 was significantly lower in the superficial lesion group than in the endometrioma group. Meanwhile, the expression of miR-10b in the deep lesion group was significantly lower than in the endometrioma group. No differences were observed in the expression of the following miRNAs: miR-9, miR-10a, and miR-204 [64].

Like Haikalis et al. (2018), our study can also lead to the conclusion that the pattern of expression of miRNAs depends on the type of endometriotic lesion analyzed. Regarding the miRNAs and genes selected herein, we observed that the superficial lesion phenotype had lower pro-survival gene expression levels and miRNAs involved in this pathway, indicating that this phenotype has a different pathophysiological mechanism in relation to deep endometriosis and ovarian endometrioma.

Our samples were not microdissected; therefore, we cannot exclude the possibility that our tissue samples contain non-endometriotic cells. Saare et al. (2014) showed significantly different gene and miRNA expression in peritoneal endometriotic lesions compared to healthy peritoneal tissues [65]. However, we believe that an endometriosis lesion is not just the ectopic endometrium, but the entire adjacent inflammatory process, so that this would also be part of the lesion. Malysheva et al. (2020) examined the expression of genes in the peritoneum of patients with endometriosis and healthy women, finding equally high level of expression of genes in endometriotic lesions and underlying peritoneum, indicating a probable common origin of these tissues [66].

The limitations of the present study include the predominance of endometriosis in advanced stages (III-IV), therefore hindering the extrapolation of our results to milder stages of the disease. Furthermore, it was not possible to obtain samples exclusively from patients who did not use hormones due to the clinical treatment currently recommended before surgery. Another confounding variable is the phase of the menstrual cycle, which was not controlled in our experiments, as samples were collected during surgery, which could interfere with our results.

Challenges for future studies include the standardization of the definitions of cases and controls, the severity of endometriosis, the phase of the menstrual cycle, tissue sampling, and laboratory methods. Such factors demand rigorous attention in the elaboration of the study design. These potentially confounding variables have been neglected and, therefore, make comparisons between studies difficult and may be responsible for the controversial results [67].

## 4. Methods and Materials

### 4.1. Sample Collection

Samples of deep lesions (*n* = 10), superficial lesions (*n* = 10), and ovarian endometriomas (*n* = 10), as well as endometrial biopsies of these respective patients, were collected from patients affected by endometriosis undergoing treatment at the Clinics Hospital of Ribeirão Preto (HCRP). Endometrial biopsies were also collected from women without endometriosis during the tubal ligation procedure, comprising the control group (*n* = 10).

The samplings were carried out in 2018 and 2019. In addition, samples from the endometriosis biorepository were also used, approved by the Research Ethics Committee of the Clinics Hospital of Ribeirão Preto (HCRP Process No. 9699/2006). To this end, we requested the integration of the biorepository with this research project, which was approved in 2006. This study was approved by the Ethics Committee of the HCRP tertiary University Hospital (Protocol No. 12514/2017). All experiments were performed in accordance with relevant guidelines and regulations.

Patients who met the following inclusion criteria were included in the study: women with endometriosis, diagnosed with any stage of the disease during a surgical procedure (according to the American Society for Reproductive Medicine), able to provide informed consent, and who were of reproductive age (between 18 and 45 years old).

Sample collection was performed during surgery, and the tissue was stored in a freezer at −80 °C in RNAlater for further gene expression analysis using the RQ-PCR technique. We reviewed the medical records of each patient, collecting data such as: date of birth, age, contraceptive method, parity, habits, the stage of the disease, medications in use, other associated diseases, and lesion characteristics. All women signed an informed consent form.

### 4.2. Extraction of Tissue RNA

The tissue samples were submitted to RNA extraction using the AllPrep DNA/RNA/miRNA Universal Kit (Qiagen, Hilden, Germany). To this end, the sample fragments were weighed as not to exceed 30 mg and homogenized using a Polytron^®^ device. From then on, the manufacturer’s protocol was followed and, immediately after RNA extraction, the samples were stored in a freezer at −80 °C. In order to verify the integrity of the obtained RNA, the samples were analyzed in a 4200 TapeStation System (Agilent Technologies, Santa Clara, CA, USA). For the quantification of the total RNA concentration, Thermo Scientific NanoDrop 2000 equipment was used (Thermo Fisher Scientific, Waltham, MA, USA).

### 4.3. Synthesis of Complementary DNA (cDNA)

For the synthesis of miRNA cDNA, we used the TaqMan™ Advanced miRNA cDNA Synthesis Kit (Applied Biosystems, Waltham, MA, USA), while for the genes, we used the High-Capacity RNA cDNA Kit (Applied Biosystems). cDNA synthesis was carried out following the manufacturer’s protocol regarding the quantities of reagents and cycle time. For miRNA, we used 10 ng, and for the genes, 100 ng of total RNA. After synthesis, the samples were stored in a freezer at −20 °C. Before the real-time PCR reaction, the following dilutions of the synthesized cDNA in DEPC water were performed: 1:10 for the miRNA and 1:4 for the genes.

### 4.4. RQ-PCR

The real-time PCR method was used to confirm the differential expression of the genes *MAPK1* (Hs01046830_m1) and *CAPN2* (Hs00965097_m1) and the microRNAs miR-30a-5p (479448_mir), miR-7-5p (483061_mir), miR-143-3p (477912_mir), and miR-93-5p (478210_mir). The choice of genes was carried out using the DAVID v6.7 (Database for Annotation, Visualization and Integrated Discovery) tool, and of miRNAs, using the miRwalk 2.0 database.

In the quantitative expression analysis, the commercially available systems TaqMan Gene Expression Assay (FAM) and TaqMan Advanced miRNA Assay (Applied Biosystems) were used for the genes and miRNA, respectively. For the miRNAs, we used the following as reference miRNA: hsa-miR-361-5p (478056_mir), hsa-miR-186-5p (477940_mir), and hsa-miR-92a-3p (477827_mir). As for the genes, *B2M* (Hs00187842_m1) and *ACTB* (Hs01060665_g1) were used as a reference.

The Real-time Quantitative Polymerase Chain Reaction (RQ-PCR) amplification reactions were performed in triplicate in a 96-well plate using the TaqMan™ Fast Advanced Master Mix reagent (Applied Biosystems), with a final volume of 10 µL. A 7500 Fast Real-Time PCR System (Applied Biosystems) was used together with the 7500 Sequence Detection System software (Applied Biosystems) to obtain the Ct values.

The standard amplification conditions were: 95 °C for 20 s, followed by 40 cycles of 95 °C for 3 s and 60 °C for 30 s (simultaneous annealing and extension). The expression data were then exported and analyzed in the Thermo Fisher Cloud Software (Last Updated 12 December 2018) Scientific, Waltham, MA, USA).

### 4.5. Statistical Analysis

In the statistical analysis, the chi-squared test was used for qualitative variables and one-way analysis of variance (ANOVA) and Tukey’s post-hoc test for quantitative variables (when there was significance in the ANOVA tests). The Mann-Whitney test was used to compare gene and miRNA expression in the eutopic endometrium of the cases and the controls, and the paired *t*-test between ectopic lesions and the eutopic endometrium of the same woman with endometriosis. The GraphPad Prism 6.0 software (GraphPad Prism, Inc, San Diego, CA, USA) was used to generate the graphs. Results were considered significant when *p* < 0.05.

## Figures and Tables

**Figure 1 ijms-24-04434-f001:**
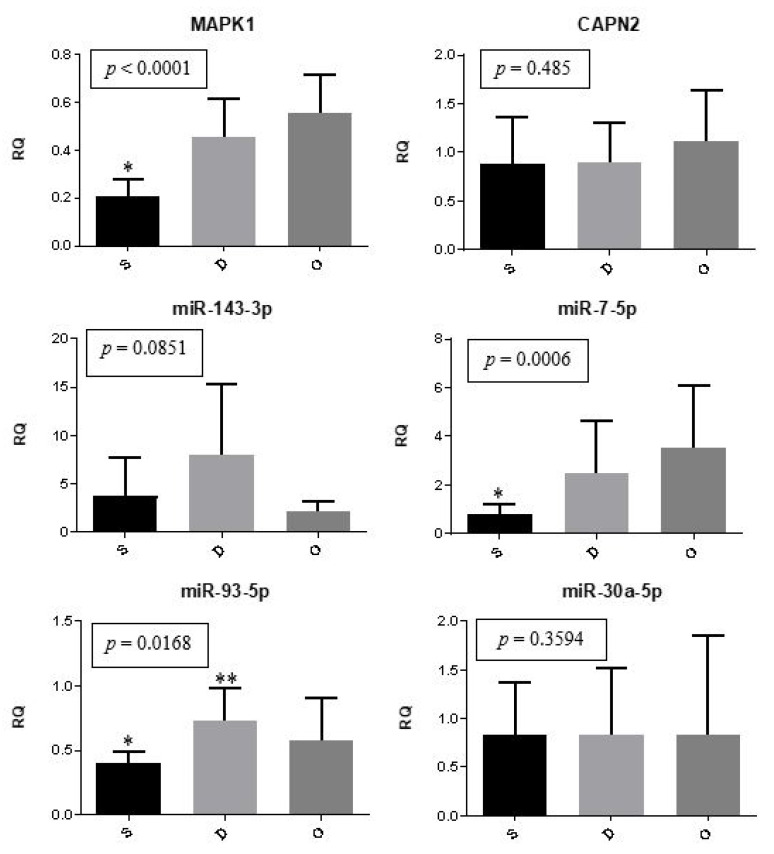
Ectopic implants. Representation of the mean (± standard error) gene and miRNA expression in superficial peritoneal endometriosis (S) (*n* = 10), deep infiltrating endometriosis (D) (*n* = 10), and ovarian endometrioma (O) (*n* = 10). The groups highlighted with * and ** showed statistically significant differences in the Tukey post-hoc test.

**Figure 2 ijms-24-04434-f002:**
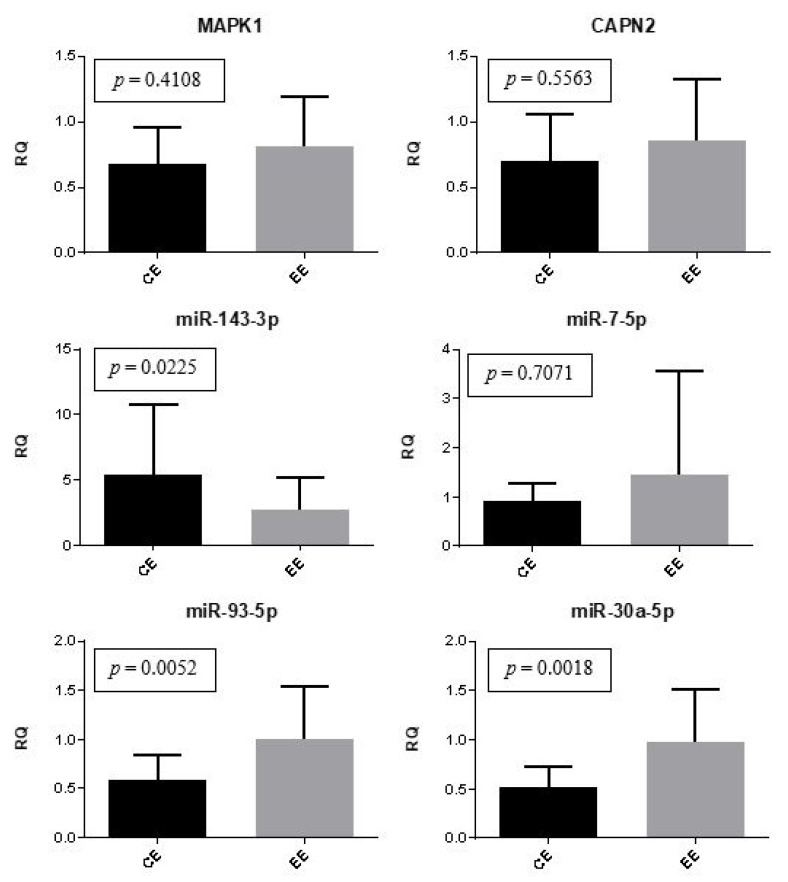
Eutopic endometrium (control versus women with endometriosis). Representation of the mean (± standard error) gene and miRNA expression in the control eutopic endometrium (CE) (*n* = 10) and the eutopic endometrium of women with endometriosis (EE) (*n* = 30). The Mann-Whitney test was used in this analysis.

**Figure 3 ijms-24-04434-f003:**
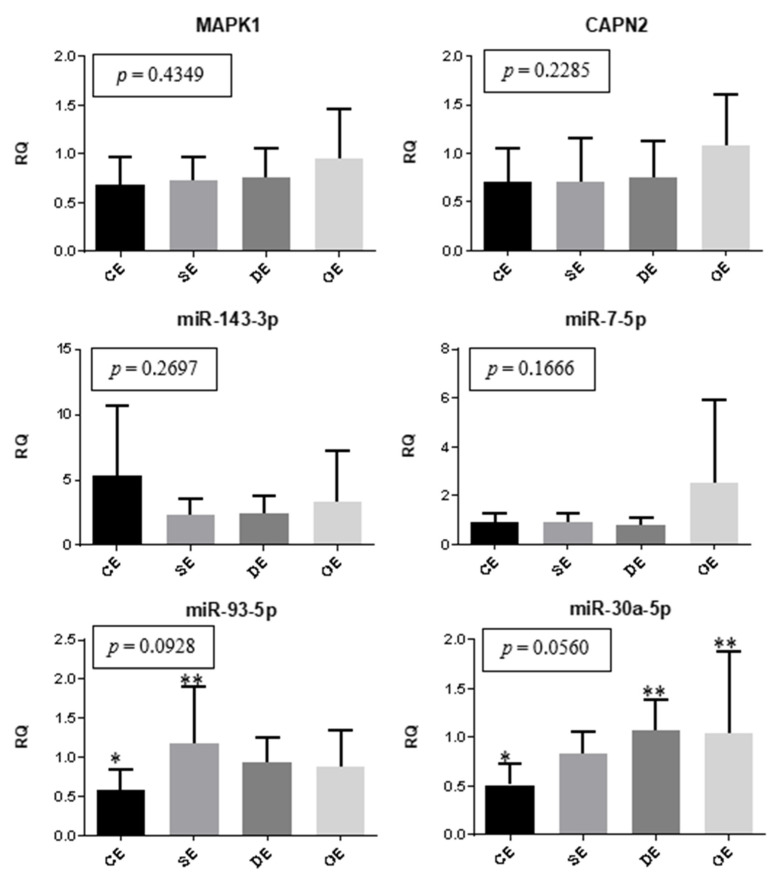
Eutopic endometrium. Representation of the mean (± standard error) gene and miRNA expression in the eutopic endometrium of women with superficial peritoneal endometriosis (SE) (*n* = 10), deep infiltrating endometriosis (DE) (*n* = 10), ovarian endometrioma (OE) (*n* = 10), and the control eutopic endometrium (CE) (*n* = 10). The groups highlighted with * and ** showed statistically significant differences in the Tukey post-hoc test.

**Figure 4 ijms-24-04434-f004:**
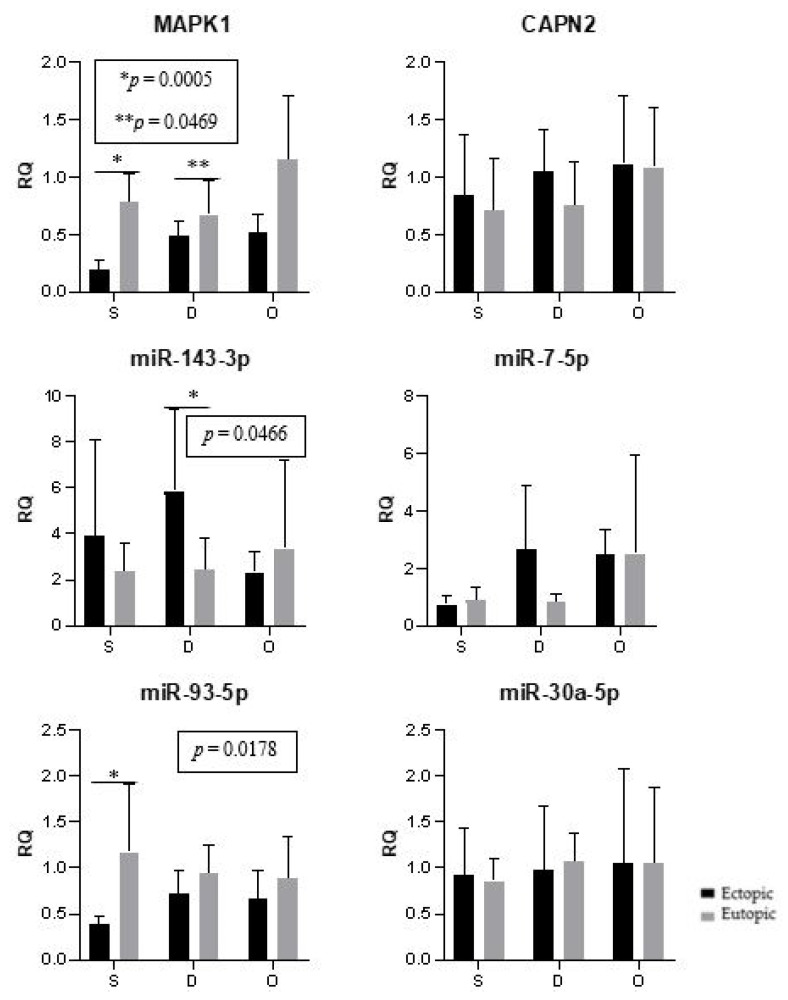
Ectopic lesions and the corresponding eutopic endometrium. Representation of the mean (± standard error) gene and miRNA expression in the ectopic lesions and the corresponding eutopic endometrium of women with endometriosis. Superficial peritoneal endometriosis (S) (*n* = 10), deep infiltrating endometriosis (D) (*n* = 10), and ovarian endometrioma (O) (*n* = 10). The groups highlighted with * and ** showed statistically significant differences according to the paired *t*-test.

**Table 1 ijms-24-04434-t001:** Demographic variables of the analyzed patients. The chi-squared test was used for qualitative variables. D: deep infiltrating endometriosis; O: ovarian endometrioma; S: superficial peritoneal endometriosis; and C: control.

		D	O	S	C	*p*-Value
Parity	Nulligravida	6	5	5	1	0.1099
	GxPx	4	5	5	9	
Medication	Present	6	6	8	3	0.1566
	Absent	4	4	2	7	
Other diseases	Present	3	7	7	3	0.0937
	Absent	7	3	3	7	
Mean age		35.3	36.1	33.2	31.2	0.1870

## Data Availability

All data generated or analyzed during this study are included in this published article and available from the corresponding author on reasonable request. RT-qPCR raw data are available upon request from the corresponding author.

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
