# Peer review of "Altered Differential Expression of Genes and microRNAs Related to Adhesion and Apoptosis Pathways in Patients with Different Phenotypes of Endometriosis"

_ijms, 2023, doi:10.3390/ijms24054434_

Round 1
Reviewer 1 Report
Dear Editor,
thank you for the opportunity to review manuscript entitled ''Altered differential expression of genes and microRNAs related to adhesion and apoptosis pathways in patients with different phenotypes of endometriosis''.
This study presents a quite good addition in overall mosaic of endometriosis pathogenesis. However, there are minor issue which need to discussed.
First of all, Introduction section is redundant and some parts have to be displaced from Introduction to Discussion section. Please cite the most recent references which reflects deteriorated quality of life in this group of patients (PMID: 35819491 and PMID: 34718292). Moreover, Discussion section has to mention possible other immunological pathways in endometriosis susceptibility, citing for example PMID: 36142815.
Author Response
We appreciate the reviewer's notes and highlight the following adjustments
- Added references 3 and 4 and the text of lines 38-42.
- We removed lines 72-74 and 77-79 and reinforced the results found for the genes of the CAPN family in the Discussion.
- We removed lines 89-108 from the introduction and partially put this information in the discussion (lines 309-311).
- About the possible other celullar pathways there is a highlight in the introduction. In the introduction, cellular processes are listed. See "escaping from immune system surveillance [6, 7]; changes in local concentrations of hormones [8] and inflammatory mediators [9]; cell adhesion [10]; tissue invasion [11]; evading apoptosis [12] ; angiogenesis [13], and ectopic cell proliferation [14]". Additionally, we emphasize that the focus of our work is apoptosis and adhesion.
Reviewer 2 Report
There are several questions.
1. The study seems to use samples collected at very different times. In our experience, different storage times for samples used to quantify RNA representation can significantly affect the results.
2. Did the samples of the endometrium of patients with endometriosis and different types of lesions used in the work belong to different women, or were these samples from the same patients?
3. It does not take into account the phase of the menstrual cycle in wich the samples were collected. This can be critically important, at least for the endometrium – if there is no reliable evidence that the expression of the studied genes and miRNA does not depend on the phase of the cycle
4. How was RNA isolated from the SPE material? These lesions are quite small, can you be shure that the material of the peritoneum or underlying connective tissue was not included in the analysis? Please describe in detail the procedure for obtaining this type of material. Was there histological confirmation of the diagnosis?
5. There is evidence that the expression of some mRNA and miRNA in SPE lesions overlaps to some extent with the expression of these RNAs in the underlyind tissue (in the peritoneum). (https://doi.org/10.1371/journal.pone.0112630; https://doi.org/10.1134/S1990519X20020066). Perhaps this explanation is also true for this work.
6. Different types of endometriosis very often coexist in one patient, and this issue is not discussed at all in the article, neither when characterizing the groups, nor when discussing the results.
7. It follows from fig.3 that the samples do not differ by ANOVA, however, the Tukey post-hoc test is applied to them, which is not quite correct.
8. In fig.4 the legen remained, probably in Portuguese )
Author Response
We appreciate the reviewer's notes and highlight the following adjustments or comments:
- The study seems to use samples collected at very different times. In our experience, different storage times for samples used to quantify RNA representation can significantly affect the results.
Currently many works use biobank materials and although the samples were collected at different times, the tissue was stored in a freezer at -80ºC in RNAlater, The tissue samples were submitted to RNA extraction in the same period and the integrity of the obtained RNA were analyzed in a 4200 TapeStation System (Agilent Technologies), with RIN values unrelated to the year of collection. Still, reference miRNA and gene were used that make corrections in the variations of the quality of the samples.
- Did the samples of the endometrium of patients with endometriosis and different types of lesions used in the work belong to different women, or were these samples from the same patients?
The samples of the endometrium of patients with endometriosis and different types of lesions used in the work belong were from the same patients. I tried to clarify better in the materials (line 139).
- It does not take into account the phase of the menstrual cycle in wich the samples were collected. This can be critically important, at least for the endometrium – if there is no reliable evidence that the expression of the studied genes and miRNA does not depend on the phase of the cycle
Samples were collected during surgery, making it difficult to control the cycle phase and other women were using hormones. We agree that this could be a confounding variable and therefore, after pointing it out, we put it in the limitations of the work (lines 437-439).
- How was RNA isolated from the SPE material? These lesions are quite small, can you be shure that the material of the peritoneum or underlying connective tissue was not included in the analysis? Please describe in detail the procedure for obtaining this type of material. Was there histological confirmation of the diagnosis?
This was a point that was much discussed when the experimental design was being prepared. We chose not to perform the microdissection for 2 reasons: for fear of impairing the quality of the samples and also because we believe that an endometriosis lesion is not just the ectopic endometrium, but the entire adjacent inflammatory process, so that this would also be part of the lesion. , so we do not insulate the tissue. However, after pointing it out, we also put it as a possible limitation of the study (424-432 lines).
- There is evidence that the expression of some mRNA and miRNA in SPE lesions overlaps to some extent with the expression of these RNAs in the underlyind tissue (in the peritoneum). (https://doi.org/10.1371/journal.pone.0112630; https://doi.org/10.1134/S1990519X20020066). Perhaps this explanation is also true for this work.
We appreciate the indication of reference and we use these to show our limitation. See question and answer 4.
- Different types of endometriosis very often coexist in one patient, and this issue is not discussed at all in the article, neither when characterizing the groups, nor when discussing the results.
We agree with the note, so we added this topic in the introduction (lines 51-56) and discussion (lines 286-290). However, in our experiments we took care to use only one type of lesion for each patient.
- It follows from fig.3 that the samples do not differ by ANOVA, however, the Tukey post-hoc test is applied to them, which is not quite correct.
Dear reviewer, we appreciate the note and reinforce that Tukey's post-hoc test for quantitative variables was performed only when there was statistical significance for the ANOVA. In view of the note, we made the following adjustment to the text (line 206):
In the statistical analysis, the chi-squared test was used for qualitative variables and one-way analysis of variance (ANOVA) and Tukey’s post-hoc test for quantitative variables (when there was significance in the ANOVA tests).
- In fig.4 the legen remained, probably in Portuguese )
Adjusted (see page 8)
